# Feeling Respected as a Person: a Qualitative Analysis of Frail Older People’s Experiences on an Acute Geriatric Ward Practicing a Comprehensive Geriatric Assessment

**DOI:** 10.3390/geriatrics4010016

**Published:** 2019-01-25

**Authors:** Theresa Westgård, Katarina Wilhelmson, Synneve Dahlin-Ivanoff, Isabelle Ottenvall Hammar

**Affiliations:** 1Department of Health and Rehabilitation, Institute of Neuroscience and Physiology, The Sahlgrenska Academy, University of Gothenburg, 40530 Gothenburg, Sweden; Katarina.wilhelmson@gu.se (K.W.); Synneve.dahlin-ivanoff@neuro.gu.se (S.D.-I.); isabelle.o-h@neuro.gu.se (I.O.H.); 2Centre of Aging and Health-AGECAP, University of Gothenburg, 40530 Gothenburg, Sweden; 3Department of Geriatrics, The Sahlgrenska University Hospital, 40530 Gothenburg, Sweden; 4Department of Occupational Therapy and Physiotherapy, The Sahlgrenska University Hospital, 40530 Gothenburg, Sweden

**Keywords:** geriatric, frail older people, person-centered care, participation, communication, understanding

## Abstract

Comprehensive geriatric assessment (CGA) practices multidimensional, interdisciplinary, and diagnostic processes as a means to identify care needs, plan care, and improve outcomes of frail older people. Conventional content analysis was used to analyze frail older people’s experiences of receiving CGA. Through a secondary analysis, interviews and transcripts were revisited in an attempt to discover the meaning behind the participants’ implied, ambiguous, and verbalized thoughts that were not illuminated in the primary study. Feeling “respected as a person” is the phenomenon participants described on a CGA acute geriatric ward, achieved by having a reciprocal relationship with the ward staff, enabling their participation in decisions when engaged in communication and understanding. However, when a person was too ill to participate, then care was person-supportive care. CGA, when delivered by staff practicing person-centered care, can keep the frail older person in focus despite them being a patient. If a person-centered care approach does not work because the person is too ill, then person-supportive care is delivered. However, when staff and/or organizational practices do not implement a person-centered care approach, this can hinder patients feeling “respected as a person”.

## 1. Introduction

Health care services are a fundamental right for all Swedish citizens [1], yet many frail older people requiring acute medical care are faced with a poorly adapted organization of services [2]. These services do not meet the needs of frail older people who present with decreased function, multi-morbidity, and chronic diseases [3,4,5]. Frailty is related to a deterioration of multiple physiological systems in old age [6]. Symptoms can range from mild to severe, may be dynamic [7], and may have a major impact on quality of life and disability [8] as they are typically associated with restricted activity and morbidity [7]. The literature remains inconsistent with an exact definition of frailty; however, a practiced consensus used in this study follows the operational criteria that: mobility, muscle strength, balance, motor processing, cognition, nutrition, endurance, and physical activity [3] are affected. Treating frail older people—A complex patient group—Is a challenge for healthcare delivery [9]. When services lack geriatric competence and are not adapted to address the frail older person requiring hospital admission, there is a risk of being marginalized [10].

Therefore frail older people require a comprehensive geriatric assessment (CGA) [11], which is person-centered in praxis if they are to achieve active aging [12] allowing for their well-being. In order to improve the care of frail older people it is important to consider how they experience receiving CGA. CGA is the coordination of a multidisciplinary team that provides multi-dimensional diagnostic and therapeutic processes which are conducted to determine the medical, mental, and functional problems of frail older people so that a coordinated and integrated plan for treatment and follow-up can be developed. By encompassing the whole person, physiological, psychological, and social aspects related to an individual’s cumulative disease status and disability are addressed. Results of implementing a CGA model have found that older patients who received the assessment in hospital had increased safety measures documented compared with acute medical wards [13] and were more likely to be alive and in their own homes at follow-up [14]. When examining the CGA team members practicing in a hospital setting, it was identified that quality geriatric care was essentially linked to the assessment process, as this proved to be an interactive, proactive, and a non-hierarchical holistic care approach [15]. CGA is designed to enable frail older people an opportunity to feel safe, trust, and involvement in decision-making while actively aging as recommended by the World Health Organization’s (WHO) framework that is based on the principles of health, participation, and security [12].

CGA is designed to include practicing person-centeredness, an approach first coined by Carl Rogers [16], which places the person at the center of their own care. People are granted a position or role that is established by another person/group of persons who ensure a facilitating environment. Once in this environment, the person receives support and is enabled so they can draw strength from their own capabilities while contributing to their care through shared decision-making, equality, and mutual respect [16,17,18]. Person-centered care involves acknowledging, valuing, and trusting, and is an approach which is considered humanistic, dignified, and morally ethical [17,18]. Person-centered care entails getting to know the unique person, allocation of power and accountability, approachability, and flexibility by organizing and integrating the care directed towards a person’s unique values base [19].

Qualitative research on patient’s experiences of receiving CGA is sparse. One of the few studies exploring frail older people’s experiences of a CGA focused on their illness and limitations related to aging [20]. Another found that inpatient CGA recipients thought that they were merely being monitored and observed, rather than being actively treated while in the hospital, and highlighted that unresolved activities of daily living (ADL) and health needs remained despite being discharged after having received intervention from a geriatrician [21]. As far as we know, the experiences of the frail older people who received a CGA on an acute geriatric medical ward have not been studied in order to understand the underlying meaning of how they experienced their care. Therefore, the aim of this study is to explore how frail older people experienced receiving CGA on an acute geriatric medical ward.

## 2. Methods 

Conventional content analysis is largely used with a study design whose aim is to discover the phenomenon [22], as in the case of this study where research is sparse, limited, and undeveloped. In the present study, a secondary analysis of qualitative data previously used for a narrative analysis was used. The primary data analysis explored how occupations and experiences of frail older people influenced their understanding of health and medical care services (forthcoming paper). The present study revisits the interviews and transcripts of the frail older people to explore how they experienced receiving CGA on an acute geriatric ward. Conventional content analysis is a method designed to objectify the process of inferences, which are deeper than the literal surface level related to the data initially uncovered about the people, social processes, and the situations experienced [23].

### 2.1. Setting and Participants

Participants selected from an acute geriatric medical ward practicing CGA were part of a randomized control trial (RCT), registered at Clinical Trials (ID: NCT02773914), and approved by the Regional Ethical Review Board in Gothenburg (EPN Gothenburg Drn: 899-15). Ten participants aged 75 or older and screened as frail using the FRESH screen [24] were selected for this study. One month after discharge from the CGA ward, eight participants from the RCT were eligible for inclusion in the qualitative study if they scored a 25 or higher on the Mini Mental State (MMS) [25]. The remaining two participants who were not part of the RCT but received the CGA were declared cognitively intact by a physician on the ward and were contacted and asked if they were willing to be interviewed and recorded after one month after discharge from the CGA ward. 

### 2.2. Data Collection

Ten individual, face-to-face interviews were conducted using open-ended questions which took place one month after discharge from the CGA ward. All participants signed informed consent forms and received both written and verbal information about the study prior to inclusion and being interviewed. All interviews were performed in the participant’s home and were carried out from June 2016–May 2018 by the first author. The interviews were recorded digitally and lasted from 21 to 63 min. The research topics explored had the potential to be sensitive, and therefore the data collection was done individually in the comfort of the participant’s home. Except for two of the participants who were interviewed at the same time as both had been admitted to the CGA ward and one of the participants who wanted their spouse to be present. The spouse contributed throughout the interview, complementing and enriching the participant’s responses. Thus, in total ten participants aged 75–95 years were included: seven women and three men, as shown by the demographics in Table 1.

### 2.3. Data Analysis

Data was analyzed using conventional content analysis, following guidelines as described by Berg [23] and Hsieh [22]. In this analysis, the inquiry process focused on how the participants experienced being hospitalized, how they experienced they were treated, what services they received and how they experienced those services, and their goals and expectations while on the CGA ward. The analysis began with the observational processing of data, where the written data was read word by word, and exact words were then highlighted which seemed to capture key thoughts or concepts. Next, the researchers used these words to write notes about the thoughts and impressions gained from the data analysis. This process continued until labels for the codes began to emerge, which were reflective of groups of related thoughts. Code were then organized and sorted into categories to give the data meaning. The success of this analysis process relies heavily on the coding process which helps researchers to organize large amounts of text into fewer categories based on their content [27]. This inductive approach required researchers to shift their analytical directions, as the search for an emerging phenomenon was not based on people, objects, events, or situations by themselves [23], but rather was about the discovery of underlying meanings and content [22,28].

The first author—bilingual in English and Swedish—interviewed, recorded, transcribed, and translated verbatim all ten interviews from Swedish to English. Distinctive to using the conventional content analysis, researchers evaded using preconceived categories in their study and rather allowed the categories to be formed and named based on the emergence of data [22]. Frequent meetings were held by the researchers to analyze the findings; this further prompted that the emergent categories, subcategories, and codes could be defined and prepared as a written report where the findings were presented and then discussed related to previous research and the consequences related to the CGA phenomenon and process. Lastly, to strengthen trustworthiness when constructing and writing the report, data in the use of quotations from the frail older people were employed to strengthen the findings.

## 3. Results

### 3.1. Respected as a Person

Participants described how they were people with resources, previous knowledge, and carried with them their wants and wishes which made up their being as a person and feeling “respected as a person”. Participants also described how they were ill patients in need of medical attention. The likelihood of feeling “respected as a person”—the core category identified in this study—explains how participants, despite being dependent on staff, experienced receiving attention while on a CGA ward. The staff’s practices and behavior promoted participants feeling that they were welcomed, cared for, and respected. Participants wanted staff that were attentive, desired a confirmation of their existence, and to be ensured that they would receive help when it was needed. These experiences made them feel safe, calm, and valued, despite being ill on a medical ward. This was able to be achieved by having a reciprocal relationship with the ward staff which included engaging in communication and understanding while maintaining contact with each other on the ward. These experiences formed the category which was achieved by attentive staff that created an atmosphere and environment making up the category *participation in decision*-making. Feeling “respected as a person” is elucidated in Figure 1, visualizing the participants’ experiences of receiving a comprehensive geriatric assessment on an acute geriatric ward. 

### 3.2. Participation in Decision

*Participation in decision* emerged as the category which had to transpire on the ward if the participant was to achieve feeling “respected as a person”. This process alluded to by the participants, focused on how their medical concerns were addressed by the ward staff who invited them to participate in decision-making related to their wants, wishes, and needs. Participation in decision-making required that the participant engaged in communication with the staff and were able to achieve an understanding of their own situation. When engaged in communication, the participants experienced favorable attention or interest from staff while sharing and receiving information on the CGA ward. Understanding was also required if the participants were to comprehend what was happening to them on the CGA ward. Understanding could also be experienced as a reciprocal relationship, which was achieved between the participants and the staff. Participants’ understanding of their situation, medical status, treatment, ward routines, discharge plans, and the way in which they processed their experienced circumstances could vary depending on how ill they were while on the ward. Furthermore, it also depended on how they experienced how they were approached by the staff working on the ward and if they were invited to participate in decision-making. Frail older people’s perceptions of participation in decision-making was expressed from four dimensions: actively participates, delegates, not able to participate, and marginalized, as shown in Figure 1. 

### 3.3. Actively Participates 

When a participant was *actively participating* in decision-making, it necessitated that they were included, which required that they understood their situation, such as medical status, treatment, ward routines, and discharge plans. It also required that they were engaged in communication with the staff so that information could be exchanged and understood, which allowed them to reach a decision together with the CGA staff. One man described how he and his wife experienced actively participating in the decision-making process prior to being discharged home from the ward.

#8 “They asked and you could tell them what you were thinking or what you needed help with they said you can have help with this and that, and we said no we want to have help with this and that. It felt like they listened to us, and then we made a decision.”

Another aspect of the actively participating experience was described by one woman who humbly described how she did not initially see or understand what the staff did. They took the time to share their knowledge, information, and concerns to improve her understanding prior to the decision-making process.

#3 “They understood that I needed additional treatment. I didn’t understand that myself. A stubborn old lady who thinks I am going to manage myself … so I had a bit of education … I got treatment and they discovered what I needed.”

### 3.4. Delegates 

When a participant was engaged in communication but did not experience that they understood the situation enough for their *participation in decision*-making, their decision was to delegate. While the participant was engaged in communication with the staff, they chose to authorize the staff on the ward to act or represent them in a situation which required that a decision be made. Thus, the decision made by the participant was deliberate, as they were informed and understood what they were consenting to by deciding not to participate. This occurred when participants felt that the staff had better knowledge and understanding of the medical situation, so they delegated the decision-making to the staff. 

When participants perceived that the staff understood, were empathetic, competent, and/or action driven, it could result in them offering suggestions and solutions to make the situation more safe or comfortable. One woman highlighted her call for help, as she communicated her concern to the staff, eluding to the importance of the staff listening skills and understanding her acute medical needs, where she delegated the responsibility to the staff.

#6 “I told them I was going to jump from the 8th floor and die and the doctor spoke with [husband] … and said we need to do something”.

### 3.5. Not Able to Participate 

When a participant was not engaged in communication and did not understand because they felt too ill, or because the staff determined that a patient was not able to participate due to being too ill, then the staff made decisions without consulting or informing the participant. This usually occurred in the early stages of the acute medical treatment, when participants were not alert or cognizant, and participants accepted this as a necessary part of their care. One woman described her experience of being very ill when she arrived to the CGA ward after a difficult experience in the emergency department, where she did not need to decide anything. 

#4 “I was a bit confused during that time, but they told me at the hospital I was lying on ward X … they took tests every morning … and I received lots of antibiotics in an IV.” 

If a participant did not experience a joint consensus with the staff founded on communication and understanding, the likelihood of participating in decision-making and feeling “respected as a person” was diminished. Understanding while dependent on a person’s resources and previous knowledge is also heavily weighed on the approach used by the other person. If a participant did not understand, and the staff did not make an effort to share their information and insight with the other person, then an ignorant, ill-informed position is likely to result. One man who acknowledged that despite not comprehending why he was admitted to the hospital, he did not ask the staff to explain his situation. 

#10 “When one is in the hospital, there are some things as a person that you don’t really understand.” 

Participants described how they could lack receiving verbal information and thus not being able to understand their situation. One woman expressed how important engagement in communication was for them and how they looked forward to experiencing that the staff would take a personal interest and give them the attention they so desired prior to discharge. 

#6 “There was supposed to be a meeting, but it never happened. The doctor at the hospital said … that we would have a meeting … but it didn’t happen … the doctor is poor at explaining possibilities. They don’t say it directly, we can’t heal that. Instead they go the side and start talking about something else … so it is just left lying there”. 

However, she reported receiving a written form of communication at discharge. This written information was not preferred if it was experienced as replacing a conversation. However, a written form of communication was welcomed by participants who wanted it as a complement clarifying the information verbally shared. #6 “A plan of care from the doctor … we got that when we were going home … what had happened, which medicine was prescribed … they were going to follow up with me at the clinic … which is good.” 

### 3.6. Marginalized 

A participant was marginalized from participation in decision-making when not engaged in communication, informed, or given the privilege to know what was going on, despite their ability to understand. When participants became aware that they were excluded or marginalized from participation in decisions, as the staff on the ward had made decisions for the participant, it could be related to their medical concerns or the organizational routines on the ward. However, participants did not want to complain or bother staff with small things, as was the case of one participant who changed room three times in four days and described how it was to share a room with a patient with dementia who woke them during the night caressing their arm. However, feeling marginalized from the organizational routines prevented them from reporting it or complaining to the staff, despite the fact that their personal space was not respected by a fellow patient. 

#9 “It was a bit strange … but we didn’t question their decision [regarding room organization]; we figured they had enough to do.”

Another marginalized episode can be illustrated when the participant experienced that the shared information they exchanged with the staff was not acknowledged or received, or when the staff gave them information but there was no opportunity to discuss. A unidirectional communication method resulted in the participants feeling that there was disengagement with the ward staff; due to a lack of information, which made them unaware of their medical status and conditions or what was expected of them. This in turn could affect future issues related to the participant’s medications, treatments, symptoms, or discharge plans. One woman expressed how sudden and impersonal the decision that was made on her behalf without her knowledge was experienced when the staff told her she had to change hospitals:

#4 “They just came in and told me I was moving … it happened very fast.”

Lastly, experiences of being marginalized occurred when an engaging conversation was neglected and resulted in the staff’s lack of understanding related to patients’ care, treatment, and medication while on the ward. One participant described how she felt that she had information that the staff did not understand or acknowledge, which caused her to be misdiagnosed and mistreated.

#2 “They filled me with very strong medicine ... even morphine for the pain … I wasn’t admitted for pain. They should have understood that something was wrong, instead of giving me loads of painkillers. I was admitted for vertigo.”

## 4. Discussion

This study is among the few to have used qualitative interviews to ascertain frail older people’s experiences of receiving a comprehensive geriatric assessment on an acute geriatric ward. Participants who felt that they were welcome and cared for by attentive staff, who confirmed their existence as a person, was what frail older people in this study described. This experience was strengthened by the person-centered approach used by staff on the CGA ward which made them feel “respected as a person”. Ekman et al. [29] found that when staff practiced person-centered care the person felt placed at the center of their clinical decisions, which considered their strengths, future plans, and rights. Person-centered care is critical for ensuring quality geriatric medical and health care services, and person-centered care is law in Sweden; Patient Act (2014:821) [30] which states that patients should be treated as people who participate in decision-making.

The positive experiences of frail older people when included to *participate in decision*-making in this study resulted in participants feeling “respected as a person”. They also experienced this when the staff practiced approaches which enabled their engagement and participation in decisions. Ringdal et al. [31] found that participation is best promoted on the ward and that it begins with the team’s understanding of the participant’s unique needs for care and their preferences, where establishing a good relationship with patients and understanding their ability to participate despite medical concerns is beneficial. The quality of interactions of older people’s experiences with ward staff explored by Bridges et al. [32] reported that experiences were shaped by how their hospital admission was perceived. This primarily was related to how they wanted the staff to interact with them so they could maintain their identity (“see who I am”), create community (“connect with me”), and share decision-making (“involve me”) [32]. The results in our study strongly concur and correlate with Bridges [32], as the framework concept development generated in this study found. Our study’s participants experienced and felt “respected as a person”; the “see who I am” felt engaged in communication and understood; the “connect with me” and included to participate in decision-making; the “involve me” when staff practiced person-centered care on the CGA ward. In this context, a previous study [31] found that it is beneficial to establish a good relationship with patients and an understanding of their ability to participate, despite their medical concerns. Similarly, Chawla et al. [33] suggested that the approaches used by staff must be nuanced and encourage patients to participate in decision-making in ways they are most comfortable. Reasons to include patients to participate reported by Eldh et al. [34] found that patients continued to experience that they were self-reliant, comprehended the situation, and maintained a sense of control when included.

In this study, frail older people’s experiences of an acute geriatric ward practicing CGA are highlighted as a complex and dynamic phenomenon, and that they did not always experience as person-centered care. Obstacles which arose hindering person-centered care from systematically occurring were identified when participants in this study experienced that when they were too ill and the staff did not involve them to participate in decisions related to their care. However, such an approach must be considered viable; as these participants were not avoided or discouraged from participating in making decisions after being informed. Rather they were limited in their cognitive capacities, restricting and preventing optimal communication and understanding, which was clinically assessed by the ward staff. Therefore, the term to describe the care they received is not a person-centered approach, but rather an alternative term of “person-supportive care” as suggested by Entwistle et al. [35]. According to Munthe et al. [36], person-centered care is intended to empower and liberate patients. However, when a person is dependent, vulnerable, or fragile, these practices are challenged. In this case, practicing shared decision-making with people meeting the criteria described by Munthe et al. [36] may have an adverse effect, which leads to disempowering, exploiting, and even harming the person it was intended to support. Therefore, it can be reasoned, as in the case of this study, that frail older people not included in decision-making due to their cognitively frail state were still cared for and supported by staff with competence and understanding, but they were not empowered to make decisions when their capabilities were limited.

In this study when participants experienced not feeling “respected as a person”, they expressed that these failures or shortcomings may have been due to the ward’s organizational structure, staff behavior, and/or staffing issues, which limited or hindered practicing person-centered care being practiced on the CGA ward. A CGA should be used to assess, treat, and plan future care with frail older people [8,37,38]. Entwistle et al. [35] found that staff must understand patient goals and capabilities, despite the fact that medical and health care policies may occasionally promote initiatives that overlook that patients are people. An earlier study warned health professionals [38] that if they do not understand patient preferences, overlook or misinterpret the person, the consequences could be as harmful as misdiagnosing a disease. McCormack et al. [19] advises person-centered care is best achieved by staff who are knowledgeable and competent in practicing the approach. However, Steward et al. [39] states that learning the practices of a person-centered care approach are challenging, time consuming, and take effort to learn the professional skills and competency required for mastering this approach. Ekman [29,40] highlights that person-centeredness requires funding and supportive government directives to secure the necessary training and organizational practices needed on the hospital ward are given priority.

Participants in this study, when experiencing that the staff did not communicate with them, did not take time to understand them, and/or did not include them in making decisions; person-centered care was not being practiced. This was most likely due to traditional practices and professional attitudes which dictated how services were provided, creating organizational barriers in health care delivery as identified in earlier studies [41,42]. By further developing and supporting the organizational practices and policies on a CGA ward, gaps could be discovered as to why patient-centered care was not always experienced by the frail older people receiving the care. Therefore, CGA could benefit from including a patient satisfaction survey, such as the Hospital Consumer Assessment of Healthcare Providers and Systems survey [43] (HCAHPS survey) or the Swedish National Patient Survey [44], as a part of their comprehensive assessment after discharge. This would allow the team practicing person-centered care to have insight and feedback about patients’ perspectives related to their care. Access to data and information identified as important to the recipients of a CGA would better enable staff to learn of their shortcomings and work towards making continuous improvements related to the quality of care they provide to frail older people. By making these survey reports public, an increase in accountability and transparency could influence other medical wards not practicing CGA and person-centered care to strive to increase patient satisfaction when treating frail older people. Therefore, measuring patient-perceived quality and experiences of health care and participation in future studies with frail older people could be useful to improve and further develop the health care system from a marginalized patient’s perspective.

### Methodological Considerations and Limitations

Employing a secondary analysis was deemed feasible as the researchers’ intent was to perform a more in-depth analysis of the findings discovered in the initial study with a subset of data from that study [45]. In the case of this study, a secondary analysis of data from the initial study was used to understand the experiences of receiving a CGA, as this appeared important but was not sufficiently concentrated on in the first analysis. Trustworthiness of qualitative studies founded on secondary data analysis may be viewed as less credible because of the relationship between the researchers and the data set [45]. However, in the case of this study, the interviews, transcriptions, and analysis of data in the initial study were performed by the first author. According to Hinds et al. [45], with this closeness to data comes benefits related to the context of the study. On the contrary, the closeness may also lead researchers to prematurely establish an understanding of the phenomenon that was present in the data set but was not the focus of the initial study [45]. Aware of these concerns, the researchers closely followed the conventional content analysis method guidelines and the phenomenon which emerged was not based on preconceptions or findings of the initial study but rather surfaced as the latent meanings and content verbalized during the interviews. Additional limitations to this study are that the sample was small, taken from a native Swedish population, which was primarily women with mild to good cognitive capacity. Future studies would benefit from using a larger sample of internationally diverse participants with less cognitive capacity and a better female-to-male ratio, so that the concept development of feeling “respected as a person” could be further explored with greater diversity when being compared with this study’s findings.

## 5. Conclusions

Receiving a comprehensive geriatric assessment when including person-centered care made the frail older patient feel “respected as a person” despite being a patient. However, if a person-centered care approach does not work because the person is too ill to participate, then care must be delivered as person-supportive care. Conversely, when a person-centered care approach is not practiced, the consequences are that patients despite a CGA model may be hindered from feeling “respected as a person”. Therefore, a CGA could benefit from the use of a patient satisfaction survey after discharge to better understand the health care gaps experienced from a patient’s perspective.

## Figures and Tables

**Figure 1 geriatrics-04-00016-f001:**
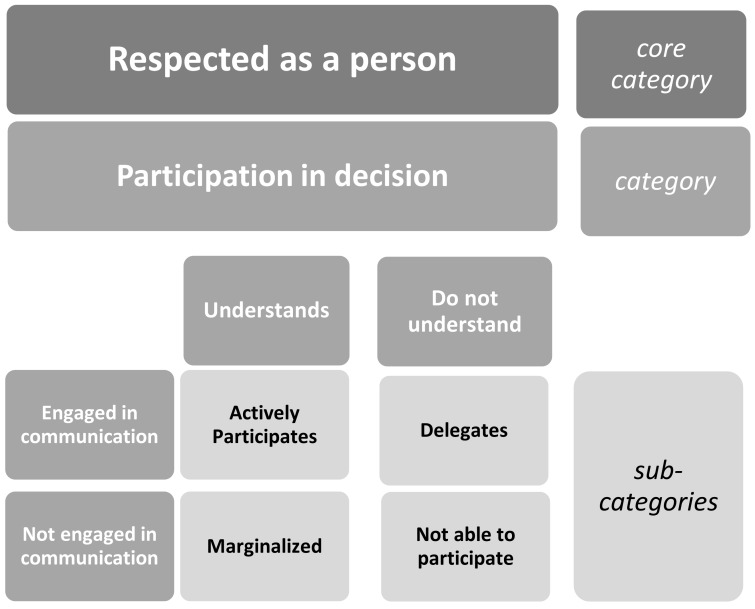
Experiences of receiving a comprehensive geriatric assessment visualized as a hierarchical process where participants felt; “respected as a person” (core category), when participation in decision (category) occurred while engaged in communication and able to understand, leading to four subcategories: actively participates, delegates, not able to participate, and marginalized.

**Table 1 geriatrics-04-00016-t001:** Demographics of participants.

Participant	Age	Gender	Self Rated Health *	Living Status
#1	88	Male	Good	Married, lives with spouse
#2	85	Female	Good	Widow, lives alone
#3	92	Female	Fair	Widow, lives alone
#4	91	Female	Good	Widow, lives alone
#5	95	Female	Good	Widow, lives alone
#6**	86	Female	Fair	Married, lives with spouse
#7	77	Female	Very good	Divorced, lives alone
#8 ***	86	Male	Good	Married, lives with spouse
#9 ***	82	Female	Good	Married, lives with spouse
#10	91	Male	Poor	Widower, lives alone

* Self rated health: a single question taken from the Short-Form Health Survey (S-36) [26], asking: “How would you rate your health: excellent, very good, good, fair or poor” at one month follow-up. ** Spouse participated. ***Interviewed jointly with spouse (both were CGA recipients).

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
