# Peer review of "Feeling Respected as a Person: a Qualitative Analysis of Frail Older People’s Experiences on an Acute Geriatric Ward Practicing a Comprehensive Geriatric Assessment"

_geriatrics, 2019, doi:10.3390/geriatrics4010016_

Round 1

Reviewer 1 Report

The manuscript describes a study of older patients’ experiences of care received on an acute-care hospital ward organized to perform person-centered CGA and associated care. The qualitative methods used involved a secondary analysis of narrative records first collected to study how earlier-life occupations and experiences affected older patients’ understanding of health and medical services. In this study, the narrative corpus was analyzed for patients’ utterances informing themes of respect for persons, participation in decisions, engagement in communications, and “grasping understanding” of their situation (see model Framework, Figure 1). While the number of subjects and the secondary (unplanned) use of the narrative materials provide a rather thin basis for the study, the paucity of prior work on patient experiences of CGA (as attested by the authors) justifies the paper, at least as a pilot for further research. The upshot here is that patient experiences of respect for persons in the CGA process appeared to be mediated by positive communications with ward caregivers, achieving an understanding of their situation, and participating in decision-making (including delay/deferral of decisional participation), with varying results depending in part on caregiver behaviors and patients’ capacities while on the geriatrics ward.

That said, the manuscript requires some significant revision. First, because narrative interpretation is so important to conveying meanings, it requires tighter, idiomatic English editing. For example, at the beginning of the paper, the authors refer to “geriatric syndrome” in the singular, which will be confusing to most readers. “Geriatric syndromes” (plural--see Sharon Inouye et al. J Am Geriatr Soc. 2007 May; 55(5): 780–791.) include delirium, falls, incontinence, physical frailty (Fried frailty), and others. Frailty, as defined by Rockwood et al., comes closest to encompassing all of these in a meta-category—the “geriatric syndrome”, if you will, although this is not discussed here. Further, while considerable narrative length should be expected in such a paper, there is redundant material here, such as reiteration in several different places of the interviews being secondarily analyzed and person-centered care being in Swedish law, and descriptive detail that does not seem to inform the salient results (e.g., Table 1). Most importantly, the emergence of the categories of person-centeredness (Figure 1) in review of the narrative materials should be sharpened and better supported descriptively. I find the “grasping understanding” and decisional participation categories thinly documented; again, one problem is idiomatic expression—does “grasping an understanding of the situation” mean the patient/person achieving an appreciation of their status, prognosis, support options, etc? Better to say so.

Second, there is much space given to review of literature having tangential relevance to the findings. It would be better to discuss how this pilot analysis should inform more direct qualitative study of geriatric patients’ CGA/associated care experiences in future research. For example, using the U.S. Medicare Hospital Consumer Assessment of Healthcare Providers and Systems Survey (HCAHPS—see wwwNaNs.gov/Medicare/Quality-Initiatives-Patient-Assessment-Instruments/HospitalQualityInits/HospitalHCAHPS.html) as one possible model, how would you modify the content to be appropriate to CGA models of hospital care and their frail elderly population? What probes would you use in interviews to get patients awaiting discharge to rate their perceptions of care integration v. fragmentation on the acute unit?...their decisional role in aftercare plans, etc?  

Future studies would be improved by conducting interviews in which patients are given discharge interviews in which their experiences of patient-centered geriatric care can be more explicitly addressed, including patient preferences for staff or family-caregivers to make decisions under given levels of patients’ real-time incapacities for participation…etc.

Author Response

Referee: 1

Comments and Suggestions for Authors

The manuscript describes a study of older patients’ experiences of care received on an acute-care hospital ward organized to perform person-centered CGA and associated care. The qualitative methods used involved a secondary analysis of narrative records first collected to study how earlier-life occupations and experiences affected older patients’ understanding of health and medical services. In this study, the narrative corpus was analyzed for patients’ utterances informing themes of respect for persons, participation in decisions, engagement in communications, and “grasping understanding” of their situation (see model Framework, Figure 1). While the number of subjects and the secondary (unplanned) use of the narrative materials provide a rather thin basis for the study, the paucity of prior work on patient experiences of CGA (as attested by the authors) justifies the paper, at least as a pilot for further research. The upshot here is that patient experiences of respect for persons in the CGA process appeared to be mediated by positive communications with ward caregivers, achieving an understanding of their situation, and participating in decision-making (including delay/deferral of decisional participation), with varying results depending in part on caregiver behaviors and patients’ capacities while on the geriatrics ward.

The authors are sincerely grateful for the very detailed feedback and analysis of our manuscript: Feeling respected as a person: a qualitative analysis of frail older people’s experiences on an acute geriatric ward practicing a Comprehensive Geriatric Assessment. Your comments were very pedagogic and the suggestions you made for further clarifying and developing the introduction, table, discussion and conclusion in the manuscript were very insightful, relevant and are addressed in detail below with clarifications of the measures taken.  

English language and style

(x) Moderate English changes required

Further, while considerable narrative length should be expected in such a paper, there is redundant material here, such as reiteration in several different places of the interviews being secondarily analyzed and person-centered care being in Swedish law,

The entire manuscript had been edited for grammar, repetitive and reiterated topics, punctuation and run on sentences.

Does the introduction provide sufficient background and include all relevant references?

(x) Can be improved

That said, the manuscript requires some significant revision. First, because narrative interpretation is so important to conveying meanings, it requires tighter, idiomatic English editing. For example, at the beginning of the paper, the authors refer to “geriatric syndrome” in the singular, which will be confusing to most readers. “Geriatric syndromes” (plural--see Sharon Inouye et al. J Am Geriatr Soc. 2007 May; 55(5): 780–791.) include delirium, falls, incontinence, physical frailty (Fried frailty), and others. Frailty, as defined by Rockwood et al., comes closest to encompassing all of these in a meta-category—the “geriatric syndrome”, if you will, although this is not discussed here.

The Introduction section of the manuscript has been rewritten after heavily weighing the reviewers’ recommendation to address and revise the definition of geriatric syndromes. However the authors chose to delete this from the text entirely, as it was not the intention of the authors to specifically address and discuss medical diseases and diagnosis as described by geriatric syndromes with regards to the participants in this study. Rather it was the intention to discuss the frail older person, and the definition as defined by Ferrucci, Fried and others.

Are the methods adequately described?

(x) Can be improved

Descriptive detail that does not seem to inform the salient results (e.g., Table 1)

The Methods section (2.0-2.3) and text have been written to more clearly describe the methods without repetitive text. Furthermore the authors felt that the demographics table gave the readers an insightfulness into who the participants were (gender and age), how they rated their own health and how they lived when not admitted to hospital, which could be useful when reading the citations in the results section. However, if the editors agree with reviewer #1, that the table is not salient then we accept it being deleted from the manuscript.

Are the results clearly presented?

(x) Can be improved

Most importantly, the emergence of the categories of person-centeredness (Figure 1) in review of the narrative materials should be sharpened and better supported descriptively. I find the “grasping understanding” and decisional participation categories thinly documented; again, one problem is idiomatic expression—does “grasping an understanding of the situation” mean the patient/person achieving an appreciation of their status, prognosis, support options, etc? Better to say so.

The Results section (3.0-3.6) was sharpened and rewritten where idiomatic language has been deleted and categories have been merged to further clarify the findings. This resulted in the emergence and labeling of new subcategories previously only discussed in the results. Furthermore this reorganization of subcategories demanded that both figure 1 and 2 in the original manuscript be scraped and reformulated as a new hierarchical process figure (now labeled as figure 1, page 5, line 206 in the track changes manuscript.(Thank you very much for this enlightening suggestion which assisted us in further developing the category system).  

Are the conclusions supported by the results?

(x) Can be improved

…there is much space given to review of literature having tangential relevance to the findings.

 The Results/Conclusion sections (4&5) has been edited and significantly reduced giving less space to the review of literature supporting the findings of the qualitative study.

 It would be better to discuss how this pilot analysis should inform more direct qualitative study of geriatric patients’ CGA/associated care experiences in future research. For example, using the U.S. Medicare Hospital Consumer Assessment of Healthcare Providers and Systems Survey (HCAHPS—see wwwNaNs.gov/Medicare/Quality-Initiatives-Patient-Assessment-Instruments/HospitalQualityInits/HospitalHCAHPS.html) as one possible model, how would you modify the content to be appropriate to CGA models of hospital care and their frail elderly population? What probes would you use in interviews to get patients awaiting discharge to rate their perceptions of care integration v. fragmentation on the acute unit?...their decisional role in aftercare plans, etc?

The discussion now reflects more clearly how CGA could improve the experiences/patient satisfaction by implementing a survey at discharge (such as HCAHPS or Swedish National Patient Survey(1)). 

 Future studies would be improved by conducting interviews in which patients are given discharge interviews in which their experiences of patient-centered geriatric care can be more explicitly addressed, including patient preferences for staff or family-caregivers to make decisions under given levels of patients’ real-time incapacities for participation…etc.

Discussion section 4 in the manuscript now introduces and discusses the use of patient satisfaction  surveys which could result in increasing the staffs’ knowledge about their shortcomings identified by the frail older person who did not experience a person-centered approach for example due to staff/organizational issues and or due to the CGA recipients “real-time incapacities preventing their participation” .

Furthermore, conclusion section 5 in the manuscript has been edited to include: the benefits of a person-centered CGA and addresses the patients’ satisfaction survey as a tool for future studies and care to improve service delivery while taking into consideration the patient’s perspective.

Lastly you have suggested a valid point of interest related to discharge interviews, experiences of “patient-centered geriatric care” preference and the use of family-caregivers in the decision making process is addressed as warranting further exploration with marginalized patients.

1.councils Ssmac. National Patient Survey [Available from: https://patientenkat.se/sv/english/.

Reviewer 2 Report

Line

Comment

Introduction

In general, the introduction does not flow well. The authors jump   from topic to topic but do not relate these topics very well to each other or   the primary purpose of this paper. Geriatric syndromes, frailty, patient   centered care, and CGA should have a thread that connects them. In addition,   the reader needs to be drawn in within the first paragraph which does not   happen. The reason this research has been undertaken should also be found or   hinted at clearly within the first paragraph.

53

Previous paragraph discussed geriatric syndromes, and introduced a   patient centered approach.  Clarify how   CGA relates to geriatric syndromes and patient-centered care.

69

This paragraph discusses person centered care and seems misplaced.   Should this perhaps go after line 52 so information on one topic is all   together?

General editing for run on sentences, misplaced punctuation and   sentence construction is advised as this muddled the meaning of significant   content, especially in the Discussion and Conclusion sections.

Is there a 2 dimensional conceptual model of the CGA  showing specifics of the relationships of   the variables? This might encompass Fig 1 and 2 within this visual model and   help to better define what CGA is, as it is not as clear in this paper as   what I have read recently.

Author Response

Referee: 2

Thank you for the feedback and analysis of our manuscript: Feeling respected as a person: a qualitative analysis of frail older people’s experiences on an acute geriatric ward practicing a Comprehensive Geriatric Assessment. Your comments and suggestions were valid and useful in assisting with the further development and improve of the manuscript with regards to both organization and clarity. Your insightful comments are addressed below in detail below, clarifying the measures taken.

Comments and Suggestions for Authors

English language and style

(x) Moderate English changes required 

General editing for run on sentences, misplaced punctuation and   sentence construction is advised as this muddled the meaning of significant   content, especially in the Discussion and Conclusion sections.

The entire manuscript had been edited for grammar, repetitive and reiterated topics, punctuation and run on sentences.

Does the introduction provide sufficient background and include all relevant references?

(x) Must be improved

Introduction

In general, the introduction does not flow well. The authors jump   from topic to topic but do not relate these topics very well to each other or   the primary purpose of this paper. Geriatric syndromes, frailty, patient-centered care, and CGA should have a thread that connects them. In addition,   the reader needs to be drawn in within the first paragraph which does not   happen. The reason this research has been undertaken should also be found or   hinted at clearly within the first paragraph.

This section of the manuscript has been rewritten after heavily weighing the reviewers’ recommendation to address and revise how geriatric syndromes and patient-centered approach related to the CGA. However the authors chose to delete “geriatric syndromes” this from the text entirely, as it was not the intention of the authors to specifically address and discuss medical diseases and diagnosis as described by geriatric syndromes with regards to the participants in this study. Rather it was the intention to discuss the frail older person, and the definition as defined by Ferrucci, Fried and others.

(Line 53) Previous paragraph discussed geriatric syndromes, and introduced a   patient centered approach.  Clarify how   CGA relates to geriatric syndromes and patient-centered care.

(Line 69) This paragraph discusses person centered care and seems misplaced.   Should this perhaps go after line 52 so information on one topic is all   together?

In addition efforts have been made to rewrite the introduction section to more invitingly lure the reader into understand why this research was undertaken relating to the CGA. Hopefully it is now less confusing with clearer definitions and explanations which previously clouded the introduction. 

Lastly the suggestion regarding (WHO) lines 52 and 69 have been compressed and reduced to avoid repetitive and redundant information that was not necessary in the manuscript

Are the methods adequately described?

(x) Can be improved

The methods section (2.0-2.3) has been written to more clearly describe the methods without repetitive text.

Are the results clearly presented?

The results section (3.0-3.6) has been reorganized, sharpened and rewritten where language has been reconstructed and deleted. While rewriting to clarify the results, this resulted in the emergence and labeling of new subcategories previously only discussed in the results. Therefore as you suggested a new 2 dimensional conceptual model has been reformulated (as a new hierarchical process figure, labeled figure 1, page 5, line 206 in the track changes manuscript). (Thank you very much for this wise and visionary suggestion which assisted us in further developing the category system and ultimately the new process figure). 

Round 2

Reviewer 1 Report

This revision is largely responsive to previous reviewer input. Patients' perspectives and appreciation of their care are certainly necessary in developing and instituting a person-centered plan of care, whether on an acute ward or over time and attendant heath and care transitions. The paper gives some insights into this.

Reviewer 2 Report

There are a few areas where semicolons are present and don't need to be.